# Thermal Stability and Hot Corrosion Performance of the AlCoCrFeNi_2.1_ High-Entropy Alloy Coating by Laser Cladding

**DOI:** 10.3390/ma16175747

**Published:** 2023-08-22

**Authors:** Li Zhang, Yan Ji, Bin Yang

**Affiliations:** Collaborative Innovation Center of Steel Technology, University of Science and Technology Beijing, Beijing 100083, China

**Keywords:** high-entropy alloy, laser cladding, thermal stability, hot corrosion

## Abstract

Both crack-free AlCoCrFeNi_2.1_ eutectic high-entropy alloy (EHEA) and Y and Hf co-doping AlCoCrFeNi_2.1_ EHEA (YHf-EHEA) coatings were prepared by laser cladding. The solidification microstructure, thermal stability, and hot corrosion performance of the coatings at 900 °C under 75% Na_2_SO_4_ + 25% NaCl molten salts were investigated. The experimental results showed that the structure of the as-deposited coatings consisted of FCC and BCC/B2 phases. After heat treatment, an Al-rich L1_2_ phase was precipitated in the FCC phase of all coatings. The grain sizes of the EHEA and YHf-EHEA coatings after heat treatment at 900 °C for 10 h increased by 27.5% and 15.7%, respectively, compared to the as-deposited coatings. Meanwhile, after hot corrosion, the spallation areas of the YHf-EHEA and EHEA coatings accounted for 14.98% and 5.67% of the total surface area, respectively. In this study, the Y and Hf co-doping did not change the microstructure morphology and phase structure of the coatings but did improve the thermal stability and resistance of the hot corrosion oxide scale spallation, providing a certain amount of data and theoretical support for the application of EHEA coatings as high-temperature protective coatings.

## 1. Introduction

Due to the presence of impurities such as Na and S in fuel, the combustion process can often lead to the deposition of a Na_2_SO_4_ salt film on the surfaces of steam pipelines and the hot-end components of gas turbines, definitely reducing the high-temperature strength and lifespan of the materials [1,2,3]. Furthermore, as human activities are increasingly expanding to coastal areas, the hot-end components in these regions face a more serious corrosive environment. The presence of NaCl particles in humid air can easily deposit on the surface of these components, accelerating the process of hot corrosion [4,5,6]. In order to mitigate or prevent hot corrosion, the preparation of protective coatings on the surface of materials is a highly economical and universally applicable method [7,8,9,10]. Currently, most researchers have aimed to investigate the hot corrosion resistance of MCrAlY (M = Fe, Ni, Co, and Pt) coatings [11,12,13]. However, there is almost no available research concerning the high-temperature hot corrosion resistance of high-entropy alloy (HEA) coatings. In the HEA system, AlCoCrFeNi_2.1_ eutectic high-entropy alloy (EHEA) with a BCC+FCC eutectic microstructure was first proposed by Lu in 2014 [14]. Here, the eutectic structure offers enhanced strength, hardness, toughness, and corrosion resistance. Therefore, EHEA has a wider range of potential applications compared to traditional alloys [15,16]. Especially in high-temperature applications, the microstructure of EHEA, with FCC+BCC phases, exhibits characteristics that are similar to the γ′ phase of nickel-based alloys, which has demonstrated outstanding resistance to oxidation [17,18]. Moreover, the unique eutectic solidification structure of EHEA provides a lower phase interface energy, and the significant sluggish diffusion effects characteristic of high-entropy alloys show its superior stability at higher temperatures [19,20]. Therefore, EHEA has great potential as a high-temperature protective coating. On the other hand, exploration of the properties of alloying elements and rare earth elements for high-entropy alloys has also never stopped [21,22,23,24]. Research about the effect of alloying elements on the performance of HEAs has predominantly concentrated on the mechanical properties while the literature indicates that conventional and high-entropy alloys and Y and Hf co-doping can effectively enhance the high-temperature oxidation resistance of alloys [25,26,27]. In high-temperature corrosive environments, the stability and hot corrosion resistance performance of HEA coatings is the most significant field of concern. Given this background, there is a need for an inclusive exploration of the thermal stability and hot corrosion performance of HEA coatings, especially EHEA coatings at high temperatures, in order to understand the microstructure and performance changes of HEA coatings.

Therefore, in order to investigate the high-temperature protective capabilities of EHEA coatings, AlCoCrFeNi_2.1_ and AlCoCrFeNi_2.1_ co-doping with Y and Hf (YHf-EHEA) coatings were prepared by laser cladding in this paper. After their preparation, the microstructure and high-temperature thermal stability of the coatings were characterized, and the hot corrosion performance of the coatings at 900 °C under 75% Na_2_SO_4_ + 25% (wt.%) NaCl mixed molten salt was investigated, which provided data support for an evaluation of the potential applications of high-entropy alloy coatings in harsh service environments such as steam pipelines and the hot-end components of gas turbines.

## 2. Materials and Methods

### 2.1. Materials

AlCoCrFeNi_2.1_ and AlCoCrFeNi_2.1_YHf gas atomized spherical powders with a size of 15–53 μm (D50 = 35.44 and 33.89 μm, respectively) and purity greater than 99.95% were used in this study. The powders’ compositions were measured using an inductively coupled plasma atomic emission spectrometer (ICP-AES, ICP-6800, Macylab Instruments Inc., Shanghai, China), as shown in Table 1. Figure 1 shows the XRD diffraction analysis results of the powders. It can be seen that both crystal structures of the powders consisted of FCC (PDF 41,505) and BCC/B2 (PDF 64,999, 184,445). However, the intensity of the B2 diffraction peak located at a diffraction angle of 31.2° was very low, which is different from that of the AlCoCrFeNi_2.1_ bulk materials while it is similar to Ding’s result [28]. It is possible to argue that the fast process does not provide sufficient time for the atomic ordering necessary for the formation or growth of the B2 structure. Therefore, the intensity of the characteristic B2 peak located at 31.2° was hard to see. The substrate was used with a 45# low carbon steel tube with a diameter of 89 mm, and the chemical composition is shown in Table 2, which was detected using a Spark Source AES (SparkCCD 6500, NCS Testing Technology Co., Ltd., Beijing, China). Meanwhile, the powder underwent a vacuum drying process at 120 °C for 3 h to eliminate moisture. This was crucial to ensure the powder’s fluidity.

### 2.2. Coating Preparation

In this study, a laser cladding manufacturing system named ZKZM-4000 (Zhongke Zhongmei Laser Technology Co., Ltd., Xian, China) was adopted for the coating preparation. The preparation parameters are presented in Table 3. Central powder feeding was used as the powder feeding method, which increased the powder utilization. In this study, argon was used as a protective and transport gas in the processing of the powders. Before the cladding process, the steel tube was polished, then cleaned with alcohol, and finally dried thoroughly, and the powder container feeds were cleaned with compressed air. In order to study the high-temperature thermal stability of the coating, vacuum heat treatment was performed at 900 °C for 10 h and water-cooled after the completion of the coating preparation, which maintained the high-temperature microstructure of the coatings.

### 2.3. Coating Characterization

The prepared coatings were cut into 10 × 10 × 5 mm specimens by wire cutting, and the microstructure of the surface and the cross-section of the coatings were observed. Before observation, the specimens were polished to 3000 grit using sandpaper, and then the specimens were polished to a mirror-like surface using 0.5 μm polishing paste. The coating microstructure was characterized by scanning electron microscopy (SEM) equipped with energy-dispersive X-ray spectroscopy (ZEISS SUPRA5 and Zeiss Genimi 450 Carl Zeiss AG, Jena, Germany). The phase structure was analyzed using an X-ray diffractometer (XRD, Smartlab Rigaku, Tokyo, Japan) with a scanning step of 2°/min and a scanning range of 25° to 85°. The coating samples for the SEM test were prepared by electrolytic corrosion using a solution of CrO_3_ (10 g) + H_2_O (90 mL), and the SEM tests were conducted at a voltage of 15 kV. The voltage and current were 5 V and 1–2 A, respectively. Electrolytic polishing samples were made with 5 vol.% perchloric acid alcohol solution, and the voltage and current were 20 V and 1–2 A, respectively.

### 2.4. Hot Corrosion Test

The hot corrosion samples with a size of 10 × 10 × 5 mm were polished. They were placed on a 120 °C heating plate, and a mixture of molten salts was evenly applied to the surface of the samples using a high-pressure spray bottle. The mixed salt content on the sample surface was maintained at 2–3 mg/cm^2^. The samples were dried and placed in alumina crucibles and then transferred to a muffle furnace at 900 °C for insulation. After 60 h of hot corrosion, the hot corrosion samples were cleaned in boiling distilled water for 30 min to remove loosely adhered surface oxides, and then they were characterized by the XRD test. The surface of the hot corrosion samples was observed by the SEM test after gold spraying, and the purpose of gold spraying was to enhance the conductivity of the corrosion test samples.

## 3. Results and Discussion

### 3.1. Microstructure of the Coatings

Figure 2 shows the XRD results of both as-deposited coatings. It can be observed that their structures consisted of FCC and BCC/B2, which are the same as the XRD results of the powders.

Figure 3 shows the cross-sectional microstructures of the EHEA and YHf-EHEA coatings. From Figure 3a,d, one can see that their thicknesses were 744 and 671 μm, respectively. No significant cracks were found on them. Figure 3b,c,e,g shows the microstructures of the top and bottom portions of the coatings, respectively. All of the coatings showed a predominantly different structure (the dark and bright regions represent the FCC and BCC/B2 phases, respectively). The microstructure morphologies of the top portion of the EHEA and YHf-EHEA coatings are presented in Figure 3b,e, which are composed of cellular dendrites. However, those at the bottom portion of both coatings exhibited columnar dendrites, as shown in Figure 3c,g. The growth of dendrites are determined by the temperature gradient (G) and cooling rate (V) at the solid–liquid interface [29]. At the bottom of the coating, the temperature gradient within the melt pool reaches its maximum value. And the solidification rate is slow, promoting the growth of planar crystals. Additionally, at the front of the solid–liquid interface, the composition is supercooled due to solute redistribution. As a result, the advancement of the straight interface is quickly interrupted, and the growth of columnar crystals begins. The columnar dendrites at the bottom portion of the coating grow along the laser build direction, which is the direction of the heat flow. As the solid–liquid interface advances toward the top of the coating, the cooling rate increases and the ratio of G/V decreases. As a result, the columnar dendrites transform into cellular dendrites. Moreover, Figure 3f displays the high-magnification backscatter electron diffraction microstructure of the YHf-EHEA coating, revealing the presence of a Re-precipitate phase at the grain boundaries, which was identified as a bright color.

Figure 4 shows the electron backscatter diffraction (EBSD) analysis of the EHEA and YHf-EHEA coatings’ surface. Figure 4a,d display the surface EBSD morphologies. It can be observed that the coating surface was composed of a mixture of coarse and fine grains. The coarse-grained region represents the heat-affected zone of the subsequent coating layer on the previous coating layer while the fine-grained region represents the remelted zone of the previous coating layer. Subsequent statistical analysis was conducted on the average size of the surface grains and the misorientation angles of the grain boundaries. It can be observed that the average grain sizes of the EHEA and YHf-EHEA coatings were 20.44 and 26.27 μm, respectively. Meanwhile, the volume fractions of the low-angle grain boundaries (LAGBs, 2° ≤ θ < 15°) in the two coatings were 20.5% and 20.2%, respectively, indicating that both coatings consisted of high-angle grain boundaries (HAGBs, θ > 15°).

### 3.2. Thermal Stability of the Coatings

Figure 5 illustrates the XRD results of the coatings after heat treatment (HT) at 900 °C, showing that the coatings consisted of FCC and BCC/B2, which is the same as the as-deposited coatings. This indicates that the microstructure of the coatings retained the as-deposited phase structures after heat treatment, but small precipitate sizes and low contents can also result in XRD detection being inconclusive. Therefore, it was necessary to further characterize the microstructure of the coatings after heat treatment.

As shown in Figure 6, it can be observed that after heat treatment, both coatings still maintained an intact BCC structure. However, small rod-shaped precipitates with widths ranging from 110 to 340 nm were found in the FCC phase. Higher temperatures cause the precipitation of the Al-rich ordered phase L1_2_ within the FCC phase of AlCoCrFeNi high-entropy alloys. Some reports have indicated that the transformation to the ordered phase occurs at temperatures of 778 ± 2 °C [30,31,32]. Al has a more negative mixing enthalpy with other elements such as Ni and Co(Al-Ni, −22 KJ/mol, Al-Co −19 KJ/mol). At 900 °C, the diffusion of Al intensified, leading to the formation of precipitate phases with other elements. Therefore, the preliminary judgment is that the rod-shaped phase that was precipitated in the FCC is the L1_2_ phase. L1_2_ and FCC exhibited a high degree of coherency. XRD could not distinguish them. The elemental analysis results of the YHf-EHEA coating are shown in Figure 7. From Figure 7a, it can be observed that the BCC phase was AlNi-rich. Moreover, one can see that the Al, Co, Cr, Fe, Ni, Y, and Hf elements in the FCC phase were uniform. However, after heat treatment, rod-like precipitates formed in the FCC phase, which were found to be Al-rich after the EDS line scan (Figure 7b,c). In conclusion, both coating samples maintained the initial BCC/B2 morphology after heat treatment and the Al-rich rod-like L1_2_ order precipitate phase was observed in the FCC.

In order to investigate the grain evolution of the coatings after heat treatment, EBSD analysis was performed on the coating surface. Figure 8a,d display the surface EBSD morphologies of the EHEA and YHf-EHEA coatings, respectively. It can be seen that the grains in the as-deposited coatings have a tendency to grow after heat treatment. From the statistical analysis of the average grain size and volume fraction of the low-angle grain boundaries (LAGBs) in both coatings after heat treatment, it can be observed that the grain sizes of the EHEA and YHf-EHEA coatings increased by 27.5% and 15.7%, respectively, compared to the as-deposited coatings. Additionally, the volume fractions of the LAGBs decreased by 13.9% and 8.3% in the EHEA and YHf-EHEA coatings, respectively. However, it can still be observed that the surface of the coating after heat treatment maintained a mixed structure of coarse and fine grains. The reason for the higher thermal stability of the YHf-EHEA coating compared to the EHEA coating can be attributed to the solid solution of reactive elements in the matrix, which enhances the sluggish diffusion effect of high-entropy alloys. Additionally, the presence of Re particles at the grain boundaries and phase interfaces can hinder grain growth (Figure 3c).

### 3.3. Hot Corrosion Behavior

The XRD patterns of both coatings after hot corrosion for 60 h are shown in Figure 9. It can be observed that the oxide layer of the coating is mainly composed of Al_2_O_3_, Cr_2_O_3_, and spinels. The PDF numbers are 26,790, 90,157, and 35,002, respectively.

After 60 h of hot etching, the coatings retained an intact form within the crucible, but some black oxide products were found at the bottom of the crucible. Figure 10 shows the surface morphologies of the coatings after 60 h of hot corrosion at 900 °C under Na_2_SO_4_ + 25% NaCl molten salts. Table 4 shows the EDS analysis of the points that are marked in yellow in Figure 10. From Figure 10a,e, it can be seen that after 60 h of hot corrosion, both coatings were covered by oxides and corrosion products. However, the YHf-EHEA coating had a lower spallation area of oxide scales on its surface compared to the EHEA coating. After hot corrosion, the mass losses for the EHEA and YHf-EHEA coatings were determined to be 33.99 and 29.05 mg/cm^2^, respectively. The spallation areas in the EHEA and YHf-EHEA coatings accounted for 14.98% and 5.67% of the total surface area, respectively. Specifically, a certain amount of warped oxides were present in the YHf-EHEA coating oxide scales that are marked in white circle in Figure 10e. The reason for the lower spallation area of the YHf-EHEA coating compared to the EHEA coating can be attributed to the uniform distribution of the Re(YHf) and stable phase structure, which is beneficial for maintaining the beneficial RE function and improving the scale adhesion [33]. The oxide scale morphologies of the coatings are shown in Figure 10c,g and the EDS analysis is shown in Table 4. It can be concluded that the oxidation corrosion products of the coatings were mainly Al-rich oxides. Additionally, there was also a certain amount of FeCr-rich oxides. Combined with the XRD results in Figure 9, it can be concluded that the corrosion product was mainly Al_2_O_3_. By analyzing the morphology and EDS of the corroded areas, it can be concluded that the elemental depletion of the EHEA coating was faster than that of the YHf coating, and trace amounts of sulfur elements were detected in the corroded areas of the EHEA. In conclusion, all coatings exhibited the same corrosion mechanism. The YHf-EHEA coating showed better resistance to the 75% Na_2_SO_4_ + 25% NaCl molten salts hot corrosion than the EHEA coating at 900 °C. The mechanism for the coatings under Na_2_SO_4_ + NaCl hot corrosion is described as follows [34,35,36,37]. Initially, chlorine reacts with the metal to form metal chlorides (MClx). However, at a temperature of 900 °C, the metal oxides (MxOy) have a lower Gibbs formation free energy than the metal chlorides, leading to the transformation of the metal chlorides into metal oxides. At the same time, the aluminum chlorides and oxides have the lowest Gibbs free energy at 900 °C, so they can form AlCl_3_ (−347.1 KJ/mol) and Al_2_O_3_ (−1295.0 KJ/mol). Among them, the Al-rich oxide (Al_2_O_3_) is the most stable. In another way, Na_2_SO_4_ decomposes into SO_3_ and O_2_, which oxidize the Al, Cr, and Fe elements to form metal oxides.
$${Cl}^{-}+O_{2}\to O^{2-}+{Cl}_{2}$$
$$M+{Cl}_{2}\to M{Cl}_{x}\left(g\right)(M=Al,Cr,Fe,etc.)$$
$$M{Cl}_{x}\left(g\right)+O_{2}\to MO_{y}+{Cl}_{2}(M=Al,Cr,Fe,etc.)$$
$$SO_{4}^{2-}\to O^{2-}+SO_{3}\to O^{2-}+S_{2}+O_{2}$$
$$2Al+SO_{3}={Al}_{2}O_{3}+\frac{1}{2}S_{2}$$
$$2Cr+SO_{3}={Cr}_{2}O_{3}+\frac{1}{2}S_{2}$$
$$Fe+O_{2}\to {Fe}_{2}O_{3}+{Cr}_{2}O_{3}\to Fe{Cr}_{2}O_{4}$$

## 4. Conclusions

In this work, AlCoCrFeNi_2.1_ EHEA and co-doping Y and Hf AlCoCrFeNi_2.1_ YHf-EHEA coatings were successfully prepared by laser cladding. The solidification microstructure, thermal stability, and hot corrosion of the coatings at 900 °C under 75% Na_2_SO_4_ + 25% NaCl molten salts were investigated.

1. The YHf co-doping did not alter the solidification microstructure and phase structure of the EHEA coating. The microstructure morphologies of the top and bottom portions of coatings were composed of cellular and columnar dendrites, respectively. And the phase structure consisted of FCC and BCC/B2.

2. After heat treatment at 900 °C for 10 h, the BCC/B2 structure was relatively stable in all coatings while the nanoscale Al-rich L1_2_ phase was precipitated in the FCC.

3. The grain sizes of the HT-EHEA and HT-YHf coatings after heat treatment at 900 °C for 10 h increased by 27.5% and 15.7%, respectively, compared to the as-deposited coatings. Additionally, the volume fractions of the LAGBs decreased by 13.9% and 8.3% in the HT-EHEA and HT-YHf coatings, respectively.

4. Al_2_O_3_ was found to be formed when the surface oxide products of the coatings were hot corroded at 900 °C for 60 h under Na_2_SO_4_ + 25% NaCl molten salts. The YHf-EHEA coating showed better resistance to scale spallation compared to the EHEA coating.

## Figures and Tables

**Figure 1 materials-16-05747-f001:**
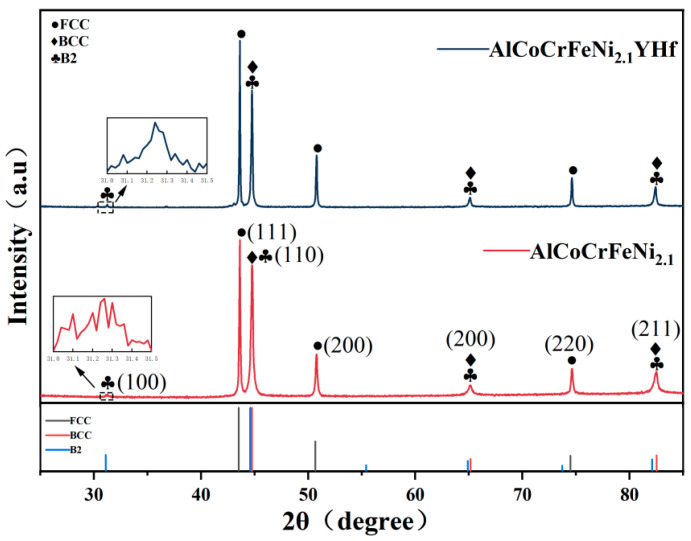
XRD patterns of the EHEA and YHf-EHEA powders.

**Figure 2 materials-16-05747-f002:**
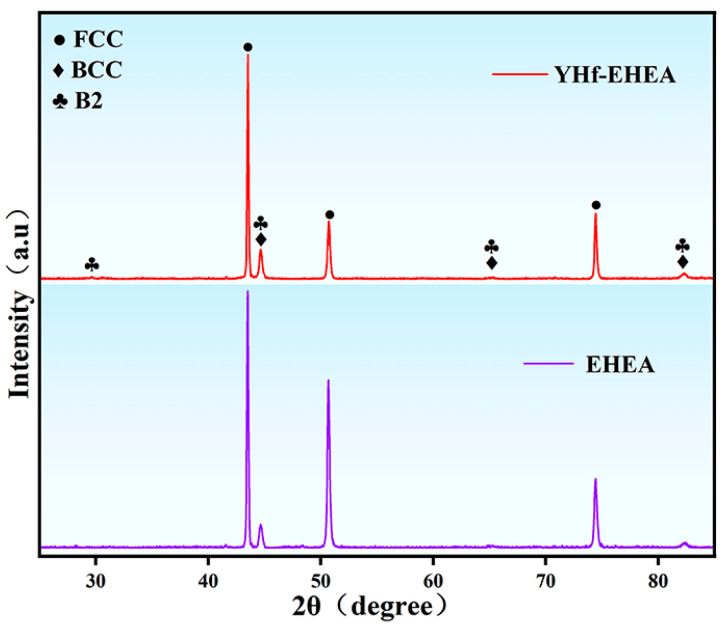
XRD patterns of the as-deposited EHEA and YHf-EHEA coatings.

**Figure 3 materials-16-05747-f003:**
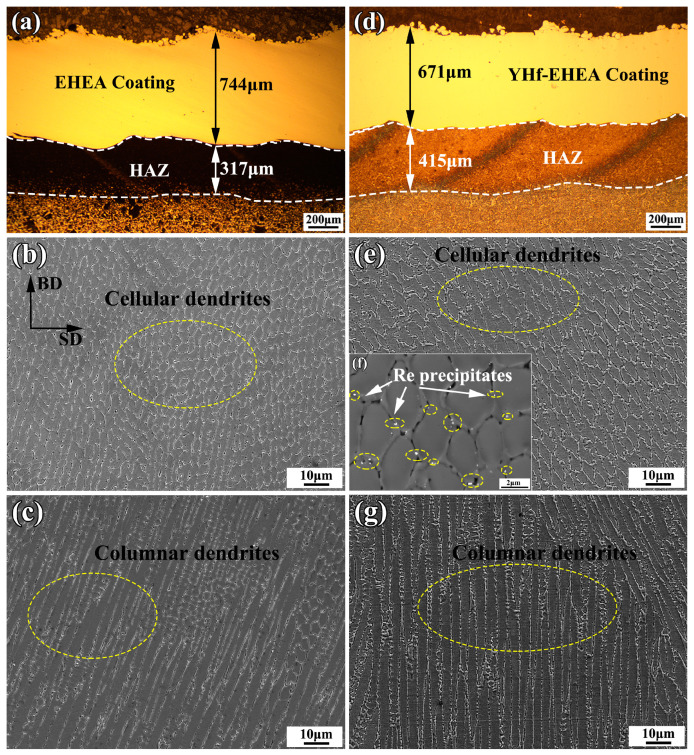
Cross-sectional microstructures of the EHEA (**a**–**c**) and YHf-EHEA (**d**,**e**,**g**) coatings; (**a**,**d**) are the cross-sectional morphologies; (**b**,**e**) and (**c**,**g**) are the microstructures of the top and bottom portions of the coating, respectively; (**f**) BSE images of the YHf-EHEA coating.

**Figure 4 materials-16-05747-f004:**
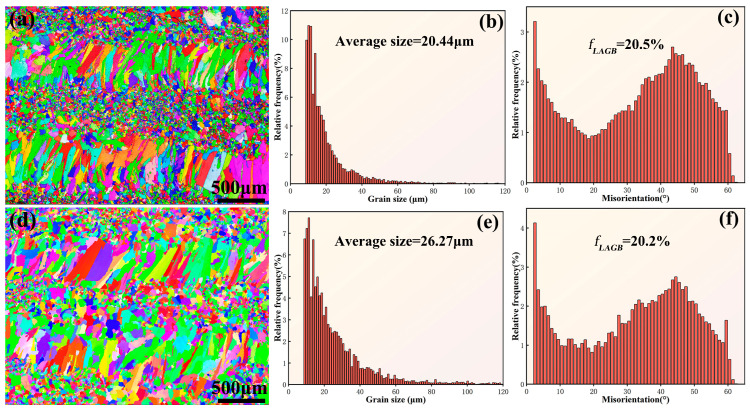
The inverse pole figure maps of the coatings’ surface, histograms of the grain size distribution, and distribution histograms of the boundary misorientation angle for the EHEA (**a**–**c**) and YHf-EHEA (**d**–**f**).

**Figure 5 materials-16-05747-f005:**
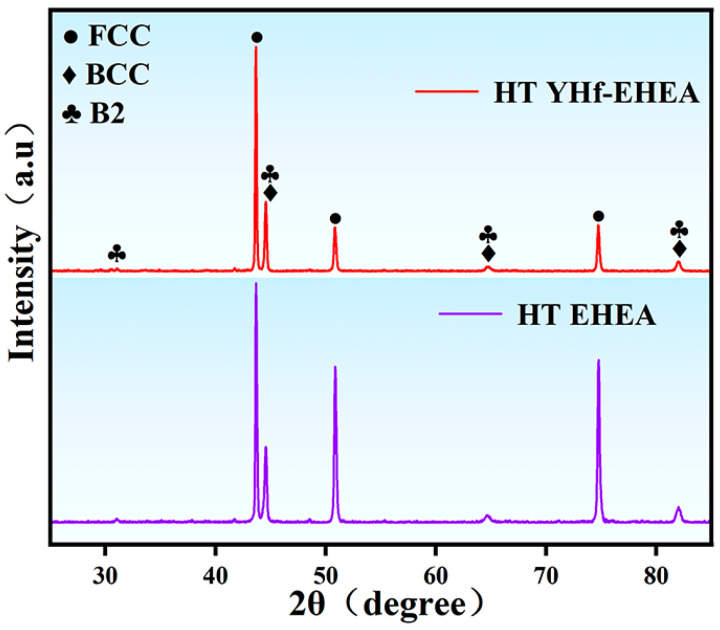
XRD patterns of the EHEA and YHf-EHEA coatings.

**Figure 6 materials-16-05747-f006:**
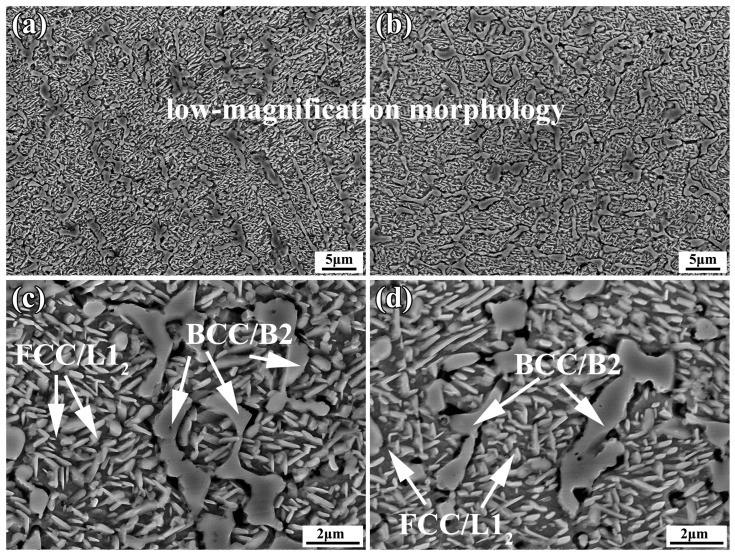
The morphologies of the EHEA (**a**,**c**) and YHf-EHEA (**b**,**d**) coatings after heat treatment; (**a**–**d**) are the low-magnification and high-magnification morphologies, respectively.

**Figure 7 materials-16-05747-f007:**
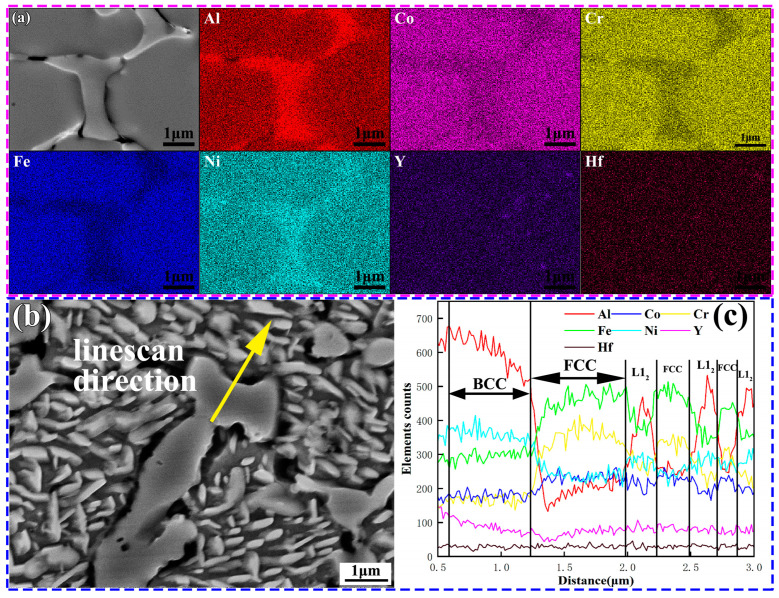
The elemental analysis of the YHf-EHEA coating: (**a**) EDS maps of the as-deposited coating and (**b**,**c**) EDS line scan direction and test results.

**Figure 8 materials-16-05747-f008:**
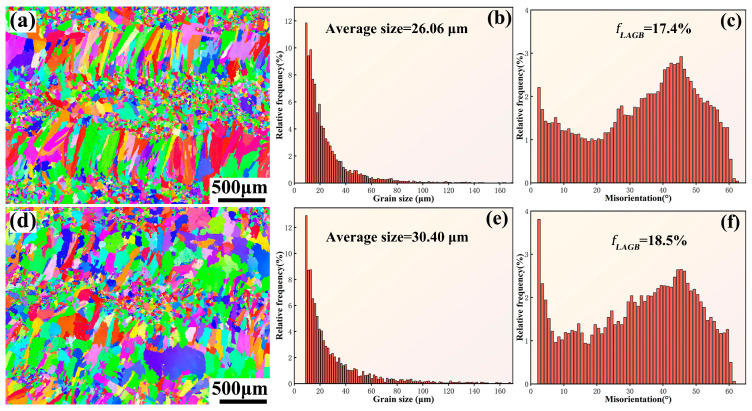
The inverse pole figure maps of the coatings’ surface after heat treatment at 900 °C for 10 h, histograms of the grain size distribution, and distribution histograms of the boundary misorientation angle for the EHEA (**a**–**c**) and YHf-EHEA (**d**–**f**).

**Figure 9 materials-16-05747-f009:**
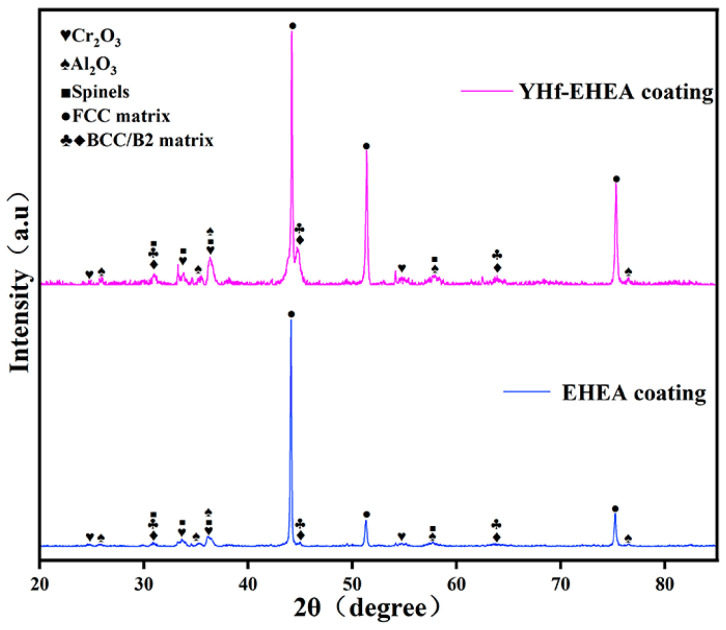
XRD patterns of the EHEA and YHf-EHEA coatings after hot corrosion for 60 h at 900 °C for 10 h under Na_2_SO_4_ + 25% NaCl molten salts.

**Figure 10 materials-16-05747-f010:**
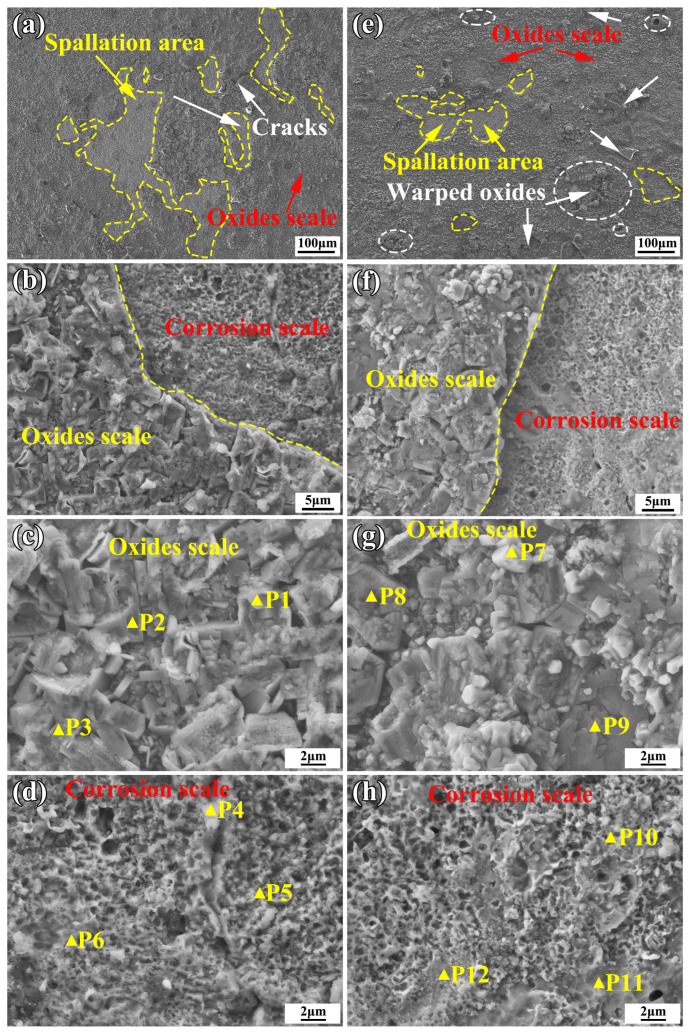
Surface morphologies of both coatings at 900 °C for 60 h under 75% Na_2_SO_4_ + 25% NaCl molten salts: (**a**–**d**) EHEA coating and (**e**–**h**) YHf-EHEA coating.

**Table 1 materials-16-05747-t001:** Chemical composition of the EHEA powders (wt.%).

Elements	Al	Co	Cr	Fe	Ni	Y	Hf	O
AlCoCrFeNi_2.1_	8.45	18.56	16.64	17.89	Bal.	-	-	0.018
AlCoCrFeNi_2.1_YHf	8.34	18.14	16.28	17.64	Bal.	0.55	0.45	0.031

**Table 2 materials-16-05747-t002:** Chemical composition of 45# steel (wt.%).

Elements	C	Si	Mn	P	S	Cr	Ni	Cu	Fe
Percent	0.44	0.27	0.60	0.018	0.007	0.017	0.009	0.009	Bal.

**Table 3 materials-16-05747-t003:** The coating preparation parameters.

Content	Parameters	Content	Parameters
Laser power	2000 (W)	Working distance	16 (mm)
Scanning speed	25 (mm/s)	Protective gas flow rate	14 (L/min)
Feeding rate	17 (g/min)	Transported gas flow rate	1.5 (L/min)
Spot diameter	2 (mm)	Overlap ratio	50%

**Table 4 materials-16-05747-t004:** The elemental compositions (at%) corresponding to the points marked in Figure 10.

Points	O	Al	Cr	Fe	Co	Ni	S
1	28.24	8.57	17.93	42.93	0	2.34	0
2	46.50	43.47	3.28	5.40	0.54	0.81	0
3	45.47	43.78	4.68	6.07	0	0	0
4	53.09	40.97	3.19	2.09	0	0.67	0
5	17.44	10.63	12.66	19.02	14.00	24.81	1.43
6	9.89	9.12	13.62	22.34	15.87	28.61	0.53
7	48.29	8.09	4.26	28.09	7.52	3.74	0
8	49.48	40.53	4.35	5.64	0	0	0
9	50.73	36.76	4.45	6.88	0.38	0.79	0
10	7.82	6.25	14.83	23.95	16.09	31.05	0
11	6.31	5.74	15.10	24.71	15.50	30.17	0
12	12.14	9.55	13.53	21.94	16.66	32.48	0

## Data Availability

The data underlying this article will be shared on reasonable request of the corresponding author.

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
