# Peer review of "Thermal Stability and Hot Corrosion Performance of the AlCoCrFeNi2.1 High-Entropy Alloy Coating by Laser Cladding"

_materials, 2023, doi:10.3390/ma16175747_

Round 1

Reviewer 1 Report

The article is devoted to the study of the properties of the high-entropy AlCoCrFeNi2.1 alloy, as well as the effect of alloying it with Y, Hf elements obtained using laser cladding methods. The key goal of this work was to establish the relationship between changes in the microstructure, thermal stability and corrosion resistance during endurance tests for stability. In general, this line of research is very promising and interesting, since these alloys are of great interest for a wide range of practical applications. The article corresponds to the subject of the declared journal and can be accepted for publication in the future after the authors answer the questions of the reviewer.

1. Authors should expand the abstract indicating new data, as well as the main results of the experiments. Also, the authors should reflect the purpose and relevance of the choice of these high-entropy alloys. At the same time, the abstract completely duplicates the last paragraph of the introduction, which is unacceptable.

2. The authors should reflect how exactly the ratios of the elements in tables 1 and 2 were determined with such a high measurement accuracy (accuracy of determination by oxygen).

3. The presence of low-intensity peaks shown in Figure 1 reflect the presence of impurity inclusions, however, the authors should reflect the numerical value of the contributions of these inclusions.

4. The study of changes in morphological features as a result of composition variation should be significantly expanded in the field of explaining the effects of the formation of a cellular structure and dendritic inclusions on the surface.

5. According to the data presented in Figure 7, the morphological features of the samples after thermal exposure clearly show the difference between the two phases and the formation of a cellular structure. In this regard, the authors should describe the mechanisms of the resulting structures as a result of external influences.

6. X-ray diffraction patterns after exposure to corrosive solutions clearly show the presence of low-intensity reflections associated with the formation of impurity inclusions, the authors should give the weight values of these inclusions in order to explain the mechanisms of their formation.

Reviewer 2 Report

The paper "Thermal stability and hot corrosion performance of the AlCo-CrFeNi2.1 high-entropy alloy coating by laser cladding" presents very interesting aspects in terms of HEA alloys. The paper can be published after some minor updates.

1. The introduction is a bit short and needs to be completed on some aspects regarding the influence of the alloying elements in terms of property usage; Suggested references: 10.4028/www.scientific.net/JERA.37.23; 10.1007/s11665-011-0014-1.

2. Reference for Table 2

3. Line 75: more parameters for SEM characterization

4. Figures 1, 2, 5, and 9 must have in the text the ICDD file compounds.

5. Line 184: Add some corrosion currents and the corrosion rate of the experimental alloys. Please add comments for the tested materials.

The rest is fine!

Reviewer 3 Report

The current manuscript was well-defined and presented. But some minor comments suggested applying on it before publishing.

1.    It is firmly suggested to review the keywords and add one more technical word. As a suggestion coating must be substituted coating by laser cladding must be substituted.

2.    An outline of the paper at the end of the introduction section is recommended. An introduction is not well defined and should be completed. 30 references are not enough in such good research.

3.    Some more important findings can be presented in the Abstract. Abstract must be edited and completed.

4.     The innovation of this paper is not highlighted. Explain clearly what the latest progress and previous work of this paper are based on?

5.    The aim of using XRD pattern in figure 1 is well explained. Did the authors expect some change in full width half maximum (FWHM) in the patterns. It is better to explain why the black pattern has a narrower graph.

6.    The other figures are well presented.

7.    Can the authors explain what is the effect of the temperature of the sample substrate on the characterization? Is this parameter measured and investigated?

Round 2

Reviewer 1 Report

The authors answered all the questions, the article can be accepted for publication.